# Effects of Concentration and Heating/Cooling Rate on Rheological Behavior of *Sesamum indicum* Seed Hydrocolloid

**DOI:** 10.3390/foods11233913

**Published:** 2022-12-04

**Authors:** Ali Rafe, Talieh Shadordizadeh, Mohammad Ali Hesarinejad, Jose M. Lorenzo, Ahmed Ali Abd El-Maksoud, Weiwei Cheng, M. R. Mozafari, Tarek Gamal Abedelmaksoud

**Affiliations:** 1Department of Food processing, Research Institute of Food Science and Technology (RIFST), Mashhad P.O. Box 91775-1163, Iran; 2Department of Food Science and Technology, Quchan Branch, Islamic Azad University, Quchan P.O. Box 83159-43919, Iran; 3Centro Tecnológico de la Carne de Galicia, Avda. Galicia n° 4, Parque Tecnológico de Galicia, San Cibrao das Viñas, 32900 Ourense, Spain; 4Área de Tecnología de los Alimentos, Facultad de Ciencias de Ourense, Universidad de Vigo, 32004 Ourense, Spain; 5Department of Dairy Science, Faculty of Agriculture, Cairo University, Giza 12613, Egypt; 6College of Food Science and Technology, Zhejiang University of Technology, Hangzhou 310014, China; 7Australasian Nanoscience and Nanotechnology Initiative (ANNI), Monash University LPO, Clayton, VI 3168, Australia; 8Food Science Department, Faculty of Agriculture, Cairo University, Giza 12613, Egypt

**Keywords:** sesame, hydrocolloids, rheological properties, concentration, temperature

## Abstract

Hydrocolloids are known as natural hydrophilic biopolymers that can contribute viscosity and gelation in solution, as well as nutritional benefits, thus, they are widely used in the food industry. In our work, hydrocolloid was isolated by aqueous extraction of *Sesamum indicum* seed at 80 °C and pH 8.0. The chemical composition and functional properties of *Sesamum indicum* seed hydrocolloid (SISH) were characterized, and the effects of concentration including 1%, 2%, and 3% as well as heating/cooling rate (1, 5, and 10 °C/min) on the rheological behavior of SISH dispersions in aqueous solution were investigated. The viscoelastic properties of SISH dispersions were characterized by small-amplitude oscillatory shear measurement. The resultant SISH consisted of 60.95% carbohydrate and 23.32% protein, and was thus endowed with a relatively high water-holding capacity, solubility, appropriate emulsifying and foaming properties. Rheological results revealed that the aqueous dispersion of SISH exhibited a non-Newtonian shear-thinning flow behavior. The viscoelastic moduli changes were found to be dependent on SISH concentration, temperature, and heating/cooling rate. Increasing SISH concentrations from 1% to 3% promoted the development of stronger cross-link network. The mechanical spectra derived from strain and frequency sweep measurements showed that the storage moduli were always higher than the loss moduli, and the loss tangent was calculated to be above 0.1 and below 1.0. Furthermore, both moduli had slight frequency dependency, and the complex viscosity exhibited an almost linear reduction with the increase of frequency. Therefore, SISH dispersion behaved as a weak gel-like system. The hysteresis of viscoelastic moduli during heating and cooling reduced with decreasing the heating-cooling rates from 10 to 1 °C/min, suggesting that SISH molecules had enough time to develop a stable and thermally irreversible network. Overall, SISH can be regarded as an acceptable hydrocolloid for generating natural food components with intriguing functional and rheological qualities in the formulation of microstructured goods.

## 1. Introduction

Hydrocolloids are macromolecules that disperse or dissolve in water and form viscous dispersions or gels [1]. They are widely recognized in changing the physical characteristics of the solution to permit gel formation or thickening, emulsification, coating, and stability [2]. Hydrocolloids made of polysaccharides have textural and functional qualities such as gel formation, stability, and control of viscosity [3], while hydrocolloids made of proteins frequently have water/fat absorption capacity, protein solubility, foaming capacity, and emulsifying activity [4,5,6]. The interactions of polysaccharide along with protein are an intriguing option to develop hydrocolloids that can improve the textural characteristics of food products. Diverse hydrocolloids have been investigated to isolate from most available natural resources, such as algae seeds, fruits, and plant exudates [7,8,9]. There have been some easy and safe procedures to produce hydrocolloids with low costs, environmental sustainability, biosecurity, biodegradability, and superior physicochemical properties [10].

Sesame (*Sesamum indicum* L.), a member of the Pedaliaceae family, is mainly cultivated in the tropical countries of Africa and Asia including Sudan, India, Afghanistan, China, and Burma, which are the main sesame-producing nations and account for roughly 60% of the world’s output. The harvested sesame seeds from its plant are traditionally considered as the queen of oilseeds due to its excellent source of edible seed and high-quality oil. Sesame seeds comprise 45–65% oil with high amounts of unsaturated fatty acids, 19–35% protein, and 14–20% carbohydrates [11], as well as low levels of moisture (3.49%) and fiber (8.22%) [12]. It has been well confirmed that sesame protein isolate from sesame meal has good functional attributes in foaming, emulsifying, gelling, and film-forming abilities [13,14,15]. Therefore, sesame seeds are considered as a good source of novel hydrocolloid made of polysaccharides and protein. A recent report has shown that sesame seed gum with 86.63% carbohydrates has great potential to be used as novel material in the production of biodegradable edible films for various biomedical and food applications [16]. Additionally, Lastra-Ripoll et al. also reported that hydrocolloid was extracted from *Sesamum indicum* seed waste under varying conditions of pH and sesame flour: water ratio, and its chemical composition, physicochemical, functional, and rheological properties were evaluated [17]. As described above, hydrocolloid can contribute viscosity and gelation in solution. Thus, it is very important to understand the rheological behavior of *Sesamum indicum* seed hydrocolloid (SISH) for its practical application as food thickeners or stabilizers. Previous reports have confirmed that concentration, temperature, and heating/cooling rate can greatly affect the rheological properties of hydrocolloids [18,19]. However, there is still no published information about the thermo-rheological behavior of SISH dispersed in water at different concentrations.

Therefore, the purpose of the current study was to investigate the effects of concentration and heating/cooling rate on dynamic rheological properties of aqueous dispersion of SISH extracted from *sesamum indicum* seed under mild conditions. The functional properties of SISH were also evaluated.

## 2. Materials and Methods

### 2.1. Materials

Sesame (*Sesamum indicum*) seeds were procured from local markets in Mashhad, Iran. The chemical reagents including ethanol (99.5% purity) and NaOH were obtained from Sigma-Aldrich (St. Louis, MO, USA). Hexane was purchased from Merck (Merck KgaA, Darmstadt, Germany). The remaining chemicals, which were employed in further processing, were all of analytical quality.

### 2.2. Extraction of SISH

The sesame seeds were initially cleaned by sieving to remove any stones and other debris. Then, the seeds were washed, dried in an oven at 60 °C, and milled using a Moulinex grinder (Model AR100, Alencon, France). The resulting sesame powder was defatted twice with hexane (hexane: powder ratio 3:1, *v*/*w*) by stirring at 250 rpm for 30 min in a lab stirrer and then centrifuging (Universal 320R, Andreas Hettich GmbH & Co. KG, Tuttlingen, Germany) at 4000× *g* for 10 min at room temperature (23 °C). The defatted sample was packaged in polyethylene bags and kept at 5 °C until further use. It was then crushed in a miller, sieved at 80-mesh size (0.177 mm, US Standard), and air-dried overnight in a hood. The extraction yield of the defatted sesame powder was about 45.08 ± 1.89% with the average particle size of 200 μm and moisture content of 10% on a wet basis.

The extraction of SISH was performed according to the previous method described by Ibañez et al. [20] with some modifications. Briefly, defatted sesame powder was mixed with water at a ratio of 1:7 at 80 °C until dispersed completely. The pH of the mixture was adjusted to 8.0 using 0.1 M NaOH solution. After that, the mixture was centrifuged for 15 min at 4000 rpm to separate the components, and the supernatant was then collected. For two hours, ethanol was diluted 1:1 with the viscous dispersion to precipitate the hydrocolloid extract. The mixture was then centrifuged, and the obtained precipitate was dried in an FDU-8624 freeze dryer (EYELA, Tokyo, Japan), milled, and stored at −18 °C.

### 2.3. Proximal Analysis

The chemical composition of SISH was determined by the methods of the Association of Official Analytical Chemists (AOAC) [21]. The samples were dried at 105 °C to assess their moisture content (AOAC, 925.10), and they were incinerated at 550 °C in a muffle furnace to evaluate their ash content (AOAC, 923.03). The fat content was assessed using the Soxhlet method (AOAC, 920.39), and the nitrogen content was determined using the macro-Kjeldal digestion method and the protein was calculated by multiplying by 6.25 (AOAC, 920.87). The total carbohydrate content was measured by subtracting other compounds form 100, and the pH of the samples was determined by a pH meter (Inolab, Weilheim, Germany). All tests were carried out in triplicate.

### 2.4. Functional Properties of Hydrocolloids

The water-holding capacity (WHC), emulsifying and foaming properties of SISH were determined according to the method described by Esmaeili et al. [5]. Briefly, the hydrocolloid dispersion (1% *w*/*v*) along with soybean oil were mixed and homogenized at 8000× *g* for 1 min. An aliquot of the emulsion was taken from the bottom of the container at 0 and 10 min after homogenization and mixed with 5 mL of sodium dodecyl sulfate solution 0.1%. The absorbance of the emulsions was measured at 500 nm with a spectrophotometer (DU730 UV/v Beckman Coulter, Pasadena, CA, USA). The absorbance at 0 and 10 min were expressed as the emulsion activity and emulsion stability index, as provided in the previous work [5].

The protein dispersion (1% *w*/*v*) was created with distilled water and pH-adjusted to 7.0 for stability and foaming capabilities. By measuring the volume of foams at 10,000× *g* immediately following homogenization for 1 min, the foaming capacity of SISH was determined and compared. The amount of foam still present after 20 min was used to gauge foaming stability. Foaming stability was considered as the foam volume remaining after 20 min and determined as follows:(1)$$FS=\frac{V_{0}\times 100}{\Delta V}$$
where *V*_0_ and Δ*V* are the initial foam volume at initial time and the alteration of foam volume during time interval 20 min, respectively.

The solubility of the SISH was assessed by dispersing 1% (*w*/*v*) of the hydrocolloids in 30 mL of distilled water stirred for 30 min. The supernatant from the centrifuged 10 mL aliquots was dried at 125 °C in a muffle until it attained a consistent weight.

### 2.5. Rheological Measurements

The SISH were gently added into distilled deionized water to achieve various concentrations of 1%, 2%, and 3%, and mixed thoroughly at 25 °C by a magnetic stirrer (Copens, Model Ms-300 Hs, Seoul, South Korea) at 360 rpm for 60 min. The dispersions were then left at 25 °C for 24 h to ensure complete hydration before the rheological measurements.

The rheological properties of SISH dispersions were analyzed by dynamic rheological measurement, which was performed on a controlled stress rheometer (Anton Paar Physica MCR 301, Stuttgart, Germany) equipped with a parallel plate geometry (50 mm diameter and 1 mm gap) and Peltier system (Viscotherm VT2, Anton Paar Companies, Darmstadt, Germany) for precise temperature control. The SISH dispersion was directly poured into the bottom plate of the rheometer at 20 °C, and the excess hydrocolloid was removed before each run. Subsequently, a thin layer of silicone oil was spread on the surface of the hydrocolloid to prevent moisture evaporation. Analysis of changes in apparent viscosity at shearing speeds ranging from 0.1 to 100 s^−1^ was done during steady-state steady-shear viscosity at 20 °C.

The Strain sweep test was performed to determine the LVE region, where dynamic storage module (G′) and loss module (G″) are independent of the strain amplitude at 20 °C and 1 Hz. The rheological parameters including G′, G″, loss tangent (tan δ), the limiting/critical value of stress (τ_c_) in the LVE region, the fracture stress and strain and crossover point were determined. It is noteworthy that the network structure broke down at the τ_c_ value, leading to a sharp reduction in the storage modulus. It can be estimated from the intersection of two asymptotic lines drawn through the initial and post-breakdown modulus data, that it can be considered to be an approximate measure of the yield stress of the material [19,22].

Frequency sweep provides suitable knowledge about the network structure. Therefore, it was carried out over a range of 0.1 to 2 Hz to investigate the viscoelastic behavior of SISH dispersion at 20 °C. The rheological characterization of SISH dispersions at different concentrations of 1, 2 and 3% (*w*/*w*) was performed by using a controlled stress rheometer. All the experiments were conducted in triplicate at 0.5% strain, which was within the LVE regime. The degree of frequency dependence of the elastic modulus is considered as an indication of the viscoelastic nature of a gel. Thus, the degrees of frequency dependence of elastic moduli (Equation (2)) were determined by the power-law model as follows [23,24].
*G*′ = *k*′*ω^q^*(2)
where *k*′ is the constant (Pa.s^q^), *q* is the dimensionless power-law index, and *ω* is oscillatory frequency (Hz).

The effect of heating/cooling rate on the rheological properties of SISH dispersion (1%) was analyzed at heating/cooling rates of 1, 5, and 10 °C/min. The SISH dispersion at 1% concentration was selected for this test, since most of the hydrocolloids were used at this level in food or pharmaceutical applications. Initially, the samples were heated up to 90 °C at the heating rates of 1, 5 and 10 °C/min at constant frequency 1 Hz and strain 0.5%, then the samples were kept at this temperature for 30 min and then cooled down to 20 °C with the same rate and kept at this temperature for at least 30 min. This procedure will enable us to evaluate the effect of heating and cooling as well as the rates on the rheological properties which may occur during food processing.

### 2.6. Statistical Analysis

Every experiment was run at least three times. Sigmaplot (version 8.0; Jandel Scientific, Corte Madera, CA, USA) and Rheoplus software (version 3.40; Anton Paar GmbH, Germany) were used to examine the rheological data and graphs, respectively. For the rheological data, one-way analysis of variance (ANOVA) was carried out. Sigmaplot was used to perform all statistical analyses at a significant level (*p* < 0.05).

## 3. Results and Discussion

### 3.1. Chemical Composition and Functional Properties of SISH

According to several previous reports on the hydrocolloid extraction from seeds [18,25], the proper removal of fatty components under alkaline conditions can help to enhance hydrocolloid yields. The extraction yield of SISH at pH 8.0 was about 12.0% on the basis of original seeds. Similar results were also observed for the hydrocolloids extracted from the seeds of *Lepidium sativum* [26] and *Plantago lanceolata* [19], in which polysaccharide molecules have hydroxyl groups that allow the formation of hydrogen bonds and proteins with negative or positive charges based on the distance between the pH and the isoelectric point (IP) causing electrostatic repulsion and hydration of charged residues, and thus promoting protein solubilization.

The chemical composition and functional properties of SISH were analyzed, and the results are shown in Table 1. The pH of the SISH solution at 1% concentration was 8.83, which was slightly higher than that of the extraction solution. It was attributed to the quantity of ethanol employed in the precipitation stage. The pH seeks to balance the pH of sesame flour and water based on its capacity to transport OH- ions when the solution is alkaline. The moisture content of the hydrocolloids was 7.76 g/100 g. The low moisture is a typical feature of hydrocolloids in exerting their functional properties when applied to the food matrix. The fat, protein, and ash were 0.47g/100 g, 23.32 g/100 g, and 7.50 g/100 g, respectively. As reported previously, the SISH have a large portion of starches and other polysaccharides due to the extraction process [17,27], which thus results in a carbohydrate content of approximately 58.42 g/100 g [28]. The high carbohydrate content in SISH can be related to the breakage of the polymeric chain of sesame polysaccharide in alkaline medium [27]. However, the protein concentration was only around 23%, which might be attributed to the ethanol remaining that combined the supernatant, and the protein could also co-precipitate with polysaccharides.

From Table 1, it was also observed that the WHC value of SISH at pH 7.0 was 4.05 g water/g, which was slightly higher than that of SISH extracted at pH 9.0 [17], and also significantly higher than that of hydrocolloids from *Pereskia bleo* leaves [29] and Tamarillo [30]. The finding could be attributed to the structure, pH, temperature, total charge density, and hydrophilicity of molecules in SISH [31]. The solubility of SISH was 34.05% at ambient temperature, which was primarily attributed to the IP of proteins. At pH levels near to the IP, the production of these conjugates significantly reduces the solubilization of SISH and may even have an impact on solubility at basic pH levels [32]. These functional properties might endow SISH with additional functionalities as stabilizers or thickeners by dispersing them, primarily in aqueous colloidal systems.

The emulsion ability and stability were 100% and 96.47%, respectively, presenting a high emulsifying capacity and enhanced stability with time. It might be explained by the high protein content in SISH, which reduced surface tension by molecular electrostatically repelling, and thus increased the stability of the emulsion [33]. The polysaccharides can also contribute increasing viscosity and improving emulsion stability. The foaming capacity and stability were observed to be 61.35% and 65.32%, respectively. The foaming ability of SISH depended on a variety of conditions, including their structure, molecular weight, protein and carbohydrate content, and the presence of other agents [34,35]. The interfacial characteristics of hydrocolloids can be directly correlated with their high protein concentration [36]. Furthermore, the polysaccharides also encouraged foaming because the polysaccharides can improve the foaming properties of protein systems by raising the viscosity of the continuous phase and forming a web of biopolymers that trap air bubbles and prevent them from collapsing or rupturing [37].

### 3.2. Rheological Properties of SISH Dispersion at Different Concentrations

#### 3.2.1. Apparent Viscosity Measurements

It is obvious that the apparent viscosity of hydrocolloid is notably influenced by the size and composition of the aggregates in hydrocolloid [38]. To investigate the effect of concentration on viscosity of SISH dispersions, the apparent viscosity of SISH dispersions at 1, 2 and 3% were evaluated against the deformation rate at 20 °C, and the results are provided in Figure 1. Regardless of SISH concentration, the viscosities of SISH dispersions almost reduced logarithmically with increasing shear rate from 0.1 to 100 s^−1^, which revealed that SISH dispersion was a Non-Newtonian fluid with a shear-thinning (pseudoplastic) behavior. The finding was consistent with aqueous dispersions of *Alyssum homolocarpum* seed gum [39], nettle seed gum [40], Chia seed mucilage [41], *Tamarindus indica* seed mucilage [42], *Plantago lanceolata* seed mucilage [19], etc. It was observed that the viscosity of SISH dispersion decreased sharply at low shear rate (0.1–1.0 s^−1^), followed by the slow decrease at high shear rate (1.0–100 s^−1^). At low shear rate, the viscosity of SISH dispersions at 3% concentration was lower than that at 1% and 2% concentration, which might state that the insoluble aggregates at 3% led to the increase of viscosity. However, as expected, when the shear rate increased, the viscosity of SISH dispersions at 3% and 2% concentration was higher than that at 1% concentration [19]. Furthermore, self-aggregation and denaturation of proteins could also enable the viscosity to return to a higher value. Similar findings were reported for the lysozyme and κ-carrageenan [38].

#### 3.2.2. Strain Amplitude Sweep Measurements

To understand the LVE region of SISH dispersion, the amplitude sweep test with raising strain for SISH dispersion at 1% concentration was carried out in controlled stress (CSR) mode at a constant frequency of 1 Hz at 20 °C. The amplitude sweep curves of SISH dispersion are displayed in Figure 2. Two different regions were observed with increasing strain, in which G′ and G″ kept almost stable in the LVE region, while G′ and G″ started to reduce in the nonlinear viscoelastic (NVE) region. Both moduli finally tended to crossover, which referred to G′ = G″ corresponding to the yield stress [22]. A critical strain (γ_0_) was defined as the strain at which G′ started to decrease sharply, and thus identified as the limit of the LVE and NVE regions. Therefore, the γ_0_ value is used as a criterion of structural strength to evaluate the deformability of the hydrocolloid. The high γ_0_ value indicates that the viscoelastic moduli are linear in a wide range of strain amplitudes, and thus confirms that the aqueous dispersion is a strong gum [43]. The γ_0_ value reduced from gels to concentrated solutions and dilute solutions, for example, <0.1% for colloidal gels, and ≥1% for natural biopolymer gels [44].

The effect of hydrocolloid concentration on the storage modulus (G′), loss modulus (G″), critical strain (γ_0_), yield stress (τ_c_), and loss tangent (tan δ) of SISH dispersions within the LVE region are summarized in Table 2. Both G′_LVE_ and G″_LVE_ values at the LVE region also increased over increasing concentrations of SISH in dispersions (Table 2). A similar trend was also observed in other hydrocolloids, gum, or mucilage [18,23], but the magnitude of the viscoelastic moduli was much lower compared to others. This might be attributed to the difference in structure and chemical composition of the viscoelastic material. γ_0_ values increased remarkably from 29.87% to 59.63% with the increasing SISH concentrations. This indicated that all SISH dispersions were identified as a gel-like structure with high stability, and the gel networks became increasingly stronger with increasing SISH levels. The corresponding stress at γ_0_ was defined as yield stress (τ_c_), that is, the strain resulting in the first nonlinear deformation in the gel structure. From Table 2, it was observed that increasing SISH concentrations led to the τ_c_ values increasing from 5.86 to 12.57 Pa. It means that the gel network got stronger [45]. Yield stress is one of the important qualitative factors for evaluating the properties of hydrocolloids in food applications, which can be artificially adjusted by changing the hydrocolloid concentration for various food applications. Loss tangent (tan δ), defined as the ratio of G″/G′, reflects the degree of energy loss relative to energy stored during deformation. A tan δ < 1 indicates predominantly solid-like (elastic) behavior, in which tan δ < 0.1 is classified as strong gel and tan δ > 0.1 is weak gel, while tan δ > 1 indicates predominantly liquid-like (viscous) behavior. As shown in Table 2, the tan δ of all SISH dispersions was in the range of 0.51–0.56, indicating the weak gel-like behavior for the obtained SISH dispersions. Furthermore, there was just a slight change in tan δ values for the aqueous dispersions with increasing concentration of SISH, indicating a weak dependence on the strain for the samples. The observation was similar with the previous reports by Hesarinejad et al. [19,23].

#### 3.2.3. Frequency Sweep Measurements

To further characterize the aqueous dispersions of SISH, the frequency sweep tests were conducted from 0.1 to 10 Hz at a fixed strain of 0.5% within the LVE region. The frequency sweep curves about the changes of the storage modulus (G′), loss modulus (G″), and complex viscosity (η *) of SISH dispersions at different concentrations (1%, 2%, and 3%) are shown in Figure 3. From Figure 3a, the magnitudes of G′ and G″ slightly increased with increasing frequencies from 0.01 to 2 Hz and had slight frequency dependency. Furthermore, the values of the storage modulus were always higher than those of the loss modulus in applied frequency range and no crossover point occurred. These observations revealed that all dispersions behaved as a weak gel (solid)-like system, which was consistent with the results of the strain sweep tests shown in Table 2. The gel-like behavior is attributed to the organization and structure of SISH. Hydrocolloids containing protein-carbohydrate conjugates are a complex system, and their mechanical characteristics depend on the interaction between molecules. Therefore, the amount of protein and carbohydrates determines the rigidity of the structure due to the chemical or physical link among molecules in a three-dimensional network, which leads to the production of stable gels (60). Furthermore, the difference between G′ and G″ values decreased with increasing frequency.

Additionally, it was also observed that the values of both storage and loss moduli did not almost change when the concentration of SISH increased from 1% to 2%, but increased remarkably when the concentration increased from 2% to 3%. Thus, the gel-like behavior was found to be dependent on SISH concentration. This finding suggested that the structure of SISH dispersion was not varied at low concentrations, while the complex and stronger structure was formed at high concentrations. As reported previously, the synchronous increase of G′ and G″ was associated with the network defects [46]. At lower concentrations, no intermolecular zone can participate in non-covalent cross junctions [47], while the intermolecular zones are formed at higher concentrations, which leads to the stronger structure of gel [48].

The complex viscosity also increased as the SISH concentration increased. This confirmed the high potential of this hydrocolloid as a good thickener or stabilizer in increasing the consistency of food systems [23]. From Figure 3b, as frequency increased, the complex dynamic viscosity (η *) fell almost linearly on a double logarithmic scale. Accordingly, similar with the above results shown in Figure 1, SISH dispersions had non-Newtonian shear-thinning behavior. Such behavior was observed in other hydrocolloids, such as xanthan gum [49], rocket seed gum [50], *Lepidium perfoliatum* seed gum [51], etc. To further understand the frequency dependency of storage moduli, the data of the frequency sweep test were fitted to power law (Equation (1)). The gel strength and structure are connected to the *q* value, with *q* ≈ 0 indicating a covalent gel and *q* > 0 indicating a physical gel. It has also been well known that elastic gel has low *q* values, and viscous gel has the *q* values close to one. Table 3 provides the values of *q* and *k*′. As expected, the *q* values were very low (0.11–0.18), and deceased with increasing SISH concentration. The frequency dependency of G′ decreased when the SISH concentration increased. It is known that *q* value is related to the strength and nature of the gel [23]. The value of the consistency coefficient (*k*′) is also an important indication of the gel strength in food processing among the rheological parameters. From Table 3, the *k*′ values increased significantly (*p* < 0.05) with increasing SISH concentrations from 1% to 3%, which indicated an increasingly strong elastic structure. This finding might be attributed to the formation of a stronger network.

#### 3.2.4. Temperature Sweep Measurements

Heating treatment can vary the rheological performance of protein-rich hydrocolloids, which may be ascribed to structural changes such as unfolding and aggregation of protein [52]. Therefore, to understand the effect of temperature on the structural and rheological property of SISH dispersion, the temperature sweep test (20–90 °C) of SISH dispersion at 1% concentration was performed with different heating/cooling rates (1, 5, and 10 °C/min) at 0.5% strain and frequency of 1 Hz. The temperature sweep curves are shown in Figure 4. Different behavior was obviously observed in the studied temperature range. During the initial heating, the value of G′ decreased gradually with the increase of temperature from 20 to 50 °C. This trend was especially pronounced for the dispersions treated at the heating rates of 5 and 10 °C/min (Figure 4b). This phenomenon about the initial reduction of G′ might be attributed to the fact that softening of the SISH network led to the increase of fluidity with increasing temperature. When the temperature was raised continuously from 50 to 90 °C and kept at 90 °C for 30 min, the values of G′ and G″ increased uniformly, indicating enhanced gel strength. The increase of G′ might be attributed to the development of a three-dimensional network structure, resulting from the intensifying interaction between proteins and carbohydrates, as well as sol–gel conversion in SISH dispersion during heating. Furthermore, the increase in G′ during heating might also be induced by the thickening effect of hydrocolloid that restricted fluid mobility. From Figure 4, it was also observed that the values of G′ and G″ nearly kept constant during cooling, indicating that cooling after heating had no significant effect on the storage and loss moduli. Although, the small changes in G′ and G″ moduli for about 20 min was observed, which does not mean that the degree of networking will not change in a longer time. The finding also implied the formation of a stable and thermally irreversible network in SISH dispersion resulting from the chemical interaction between proteins and carbohydrates in SISH. The hysteresis in the viscoelastic moduli during heating and cooling decreased with decreasing heating–cooling rates from 10 to 1 °C/min, which suggested that the molecules in SISH had enough time to form chemical bonds and to develop a stable and thermally irreversible network (Figure 4).

## 4. Conclusions

In this study, hydrocolloids with significant amounts of protein and carbohydrates were extracted from *Sesamum indicum* seed in alkaline medium (pH = 8.0). The obtained SISH had a relatively high WHC, solubility, and emulsion stability, as well as good foaming ability. The dynamic flow properties of SISH dispersed in water within the LVE region were investigated as a function of concentration (1%, 2%, and 3%) and heating/cooling rate (1, 5, 10 °C/min). Apparent viscosity measurement confirmed the aqueous dispersions of SISH, a non-Newtonian shear-thinning flow behavior at various concentrations. The storage and loss moduli were found to be dependent on the SISH concentration. Increasing in SISH concentration could give positive effects on the storage and loss moduli, as well as critical strain and yield stress. The storage moduli were always higher than the loss moduli, and the tan δ was calculated to be above 0.1 and below 1.0, which indicated that all SISH dispersions behaved as a week gel-like system. Frequency sweep tests revealed that the viscoelastic moduli of all prepared SISH dispersions showed a slight frequency dependency, suggesting that it was a cross-link network. The complex viscosity exhibited an almost linear reduction with the increase of frequency. The hysteresis of viscoelastic moduli during heating and cooling reduced with decreasing heating–cooling rates from 10 to 1 °C/min, suggesting that SISH molecules had enough time to develop a firm network form, a stable and thermally irreversible network. These results provide the valuable knowledge for potential use of novel hydrocolloid extracted from *Sesamum indicum* seed as a good thickener or stabilizer in the food industry such as meat analogues, cakes and confectionary products.

## Figures and Tables

**Figure 1 foods-11-03913-f001:**
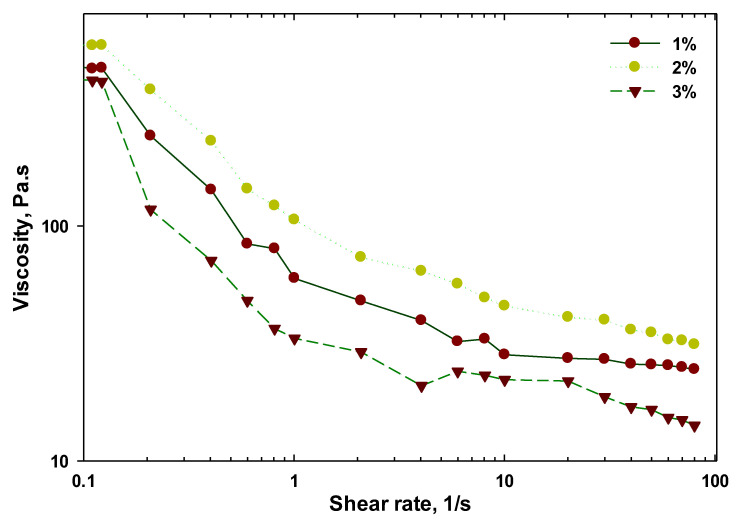
Effect of SISH concentration on the apparent viscosity of SISH dispersions as a function of shear rate from 0.1 to 100 s^−1^ at 20 °C.

**Figure 2 foods-11-03913-f002:**
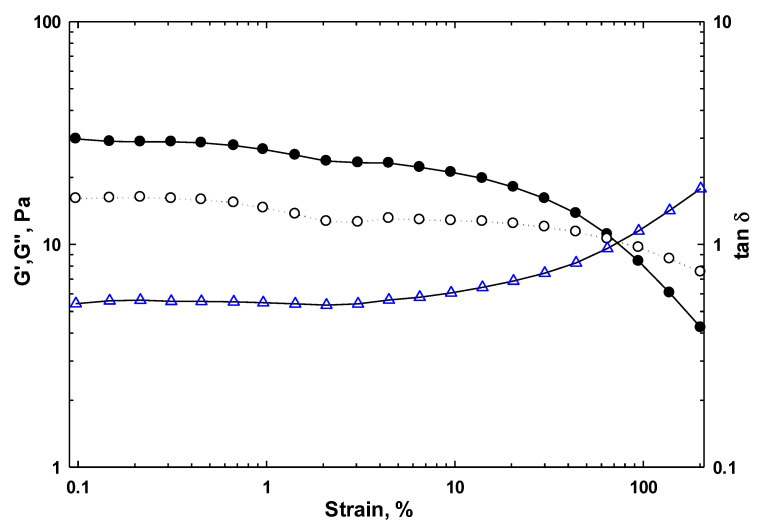
Storage modulus (G′, solid symbol), loss modulus (G″, blank symbol), and loss tangent (tan δ, blue symbol) of SISH dispersions at 1% concentration determined by strain amplitude sweep tests as a function of strain amplitude from 0.1% to 150% at a constant frequency of 1.0 Hz.

**Figure 3 foods-11-03913-f003:**
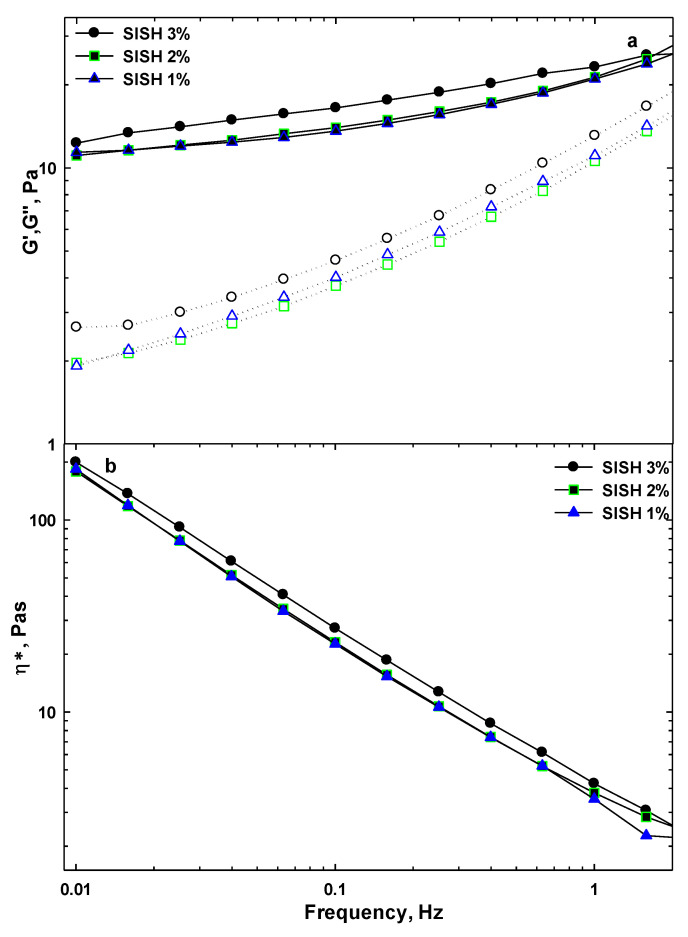
Viscoelastic modulus (**a**) and complex viscosity (**b**) of SISH dispersions at different concentrations (1, 2, and 3%) determined by frequency sweep tests from 0.01 to 2 Hz with the strain amplitude of 0.5%.

**Figure 4 foods-11-03913-f004:**
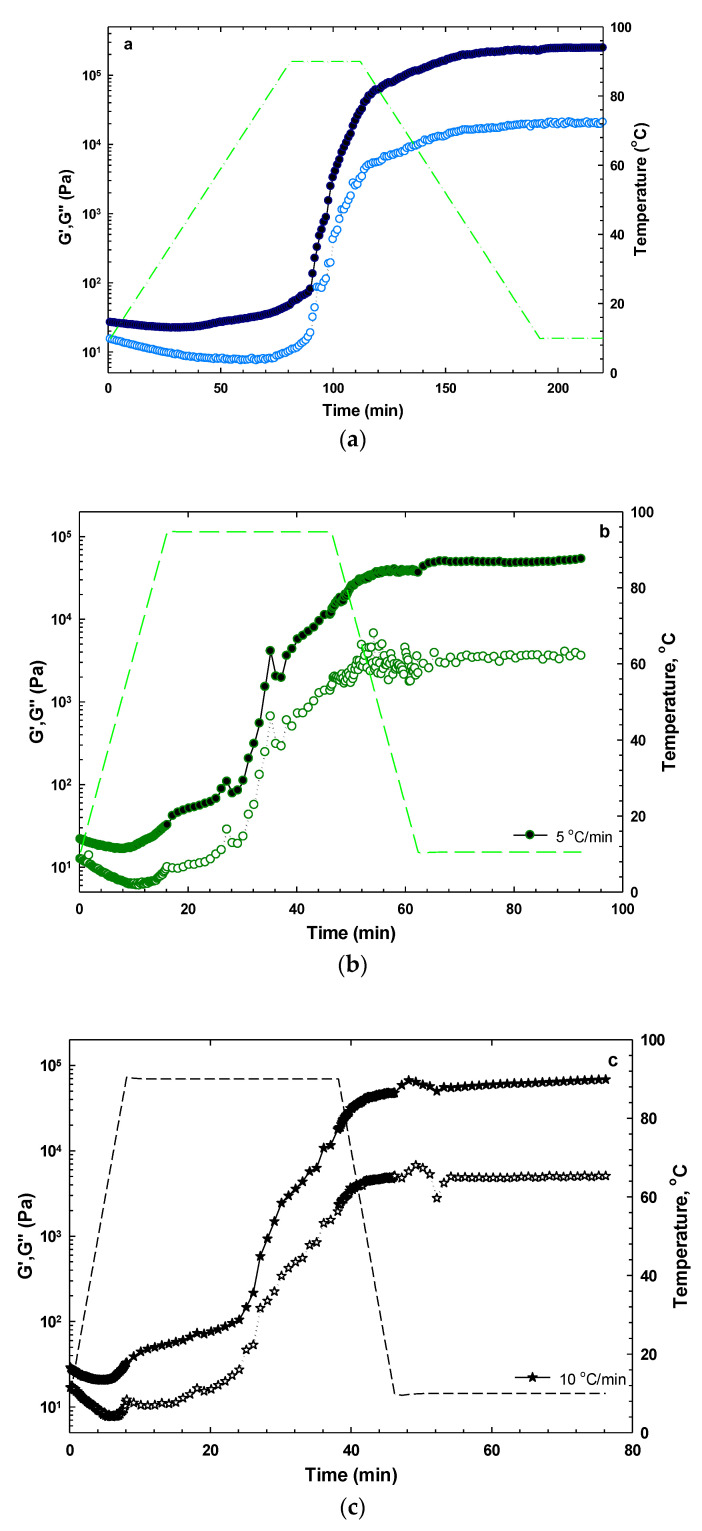
Effect of heating/cooling rate on the G′ (filled symbols) and G″ (blank symbols) of SISH dispersions at 1% concentration: (**a**) 1 °C/min, (**b**) 5 °C/min, and (**c**) 10 °C/min. All measures were conducted at the frequency of 1 Hz with the strain amplitude of 0.5%.

**Table 1 foods-11-03913-t001:** Chemical composition and functional properties of SISH.

	Parameter	Value (%)
Chemical composition	Moisture	7.76 ± 0.14
	Fat	0.47 ± 0.04
	Protein	23.32 ± 0.67
	Carbohydrate	60.95 ± 1.28
	Ash	7.50 ± 0.70
Functional properties	WHC	405.05 ± 27.38
	Emulsion ability	100 ± 0.00
	Emulsion stability	96.47 ± 0.75
	Solubility	34.05 ± 3.45
	Foaming ability	61.35 ± 1.60
	Foaming stability	65.32 ± 2.31

Data represent mean ± SD (n = 3).

**Table 2 foods-11-03913-t002:** Viscoelastic parameters calculated by strain sweep test at 1.0 Hz for SISH dispersions at different concentrations (1, 2, and 3%).

SISH Concentration (%)	G′_LVE_ (Pa)	G″_LVE_ (Pa)	Tan δ_LVE_	τ_c_ (Pa)	γ_0_ (%)
1.0	23.96 ± 0.58 ^c^	12.93 ± 0.04 ^c^	0.51 ± 0.01 ^a^	5.86 ± 0.97 ^c^	29.87 ± 1.48 ^c^
2.0	25.91 ± 0.27 ^b^	14.93 ± 0.03 ^b^	0.54 ± 0.02 ^a^	8.43 ± 1.07 ^b^	43.79 ± 2.34 ^b^
3.0	28.15 ± 0.62 ^a^	16.73 ± 0.02 ^a^	0.56 ± 0.01 ^a^	12.57 ± 1.12 ^a^	59.63 ± 3.57 ^a^

Data represent mean ± SD (n = 3). Different letters within the same column indicate the statistically significant difference between groups at *p* < 0.05.

**Table 3 foods-11-03913-t003:** Fitted power-law parameters for G′ of SISH dispersions at different concentrations.

SISH Concentration (%)	*q*	*k*′	*R* ^2^
1.0	0.18 ± 0.02 ^c^	39.56 ± 0.79 ^c^	0.96
2.0	0.14 ± 0.01 ^b^	61.85 ± 0.84 ^b^	0.90
3.0	0.11 ± 0.02 ^a^	82.72 ± 1.09 ^a^	0.91

Data represent mean ± SD (n = 3). Different letters within the same column indicate the statistically significant difference between groups at *p* < 0.05.

## Data Availability

Data are contained within the article. All data generated or analyzed during this study are included in this article.

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
