# Peer review of "Effects of Concentration and Heating/Cooling Rate on Rheological Behavior of Sesamum indicum Seed Hydrocolloid"

_foods, 2022, doi:10.3390/foods11233913_

Round 1
Reviewer 1 Report
In general, the manuscript is adequately written. Some issues to be revised by the authors:
I suggest modifying the title in section 2.3. to: “2.3. Proximal analysis”. If methodologies were done according to AOAC, then, the number of each method should be included.
Please clarify why did you use SISH concentrations of up to 3%.
Please delete the dot in line 271.
In conclusion, line 407 states: “The obtained 407 SISH had high WHC, solubility, and emulsion stability, as well as good foaming ability.”, however, how much is high? Compared to? How could you conclude that that value is actually high?
It is not clear the industrial application of this product. It is mentioned “as good thickener or stabilizer in the food industry”, but could you provide specific examples?
Author Response
C1: I suggest modifying the title in section 2.3. to: “2.3. Proximal analysis”. If methodologies were done according to AOAC, then, the number of each method should be included.
R1: Thanks a lot. It was corrected.
C2: Please clarify why you used SISH concentrations of up to 3%.
R2: Thank you for attention to our work. Hydrocolloids are generally used at low concentrations in the semi-dilute region of concentration in food application. Therefore, we used the concentrations commonly used for other hydrocolloids utilized in food formulation. The low region of concentration of hydrocolloids, lower than 0.1% used for measuring the intrinsic properties of hydrocolloids, and higher region of concentration, more than 3% makes the hydrocolloids insoluble and difficult in solvation; as the hydrocolloids molecules absorb much water.
C3: Please delete the dot in line 271.
R3: Thank you for high accuracy. It was corrected.
C4: In conclusion, line 407 states: “The obtained SISH had high WHC, solubility, and emulsion stability, as well as good foaming ability.”, however, how much is high? Compared to? How could you conclude that value is actually high?
R4: Thanks a lot. We have explained the value of the WHC, solubility, and emulsion stability in the table 1 as well as results and compare with other hydrocolloids. Based on the literature and comparison with plant hydrocolloids, we explain that the functional properties are relatively high. Thus, we changed the statement to “The obtained SISH had a relatively high WHC, solubility, and emulsion stability, as well as good foaming ability”.
C5: It is not clear the industrial application of this product. It is mentioned “as good thickener or stabilizer in the food industry”, but could you provide specific examples?
R5: Thanks for your interest. We are currently working on the application of sesame protein in meat analogues which has interesting application in Food industry, however, the sesame hydrocolloids is used in preparing some Halawa and applied in the cake and confectionary products.
Reviewer 2 Report
The paper “Effects of Concentration and Heating/cooling Rate on Rheological Behavior of Sesamum Indicum Seed Hydrocolloid” reports on the chemical and rheological characterization of an extract obtained from sesamum indicum seeds.
The topic is neither particularly interesting nor innovative, different polysaccharides and proteins can be extracted from many vegetable seeds and in the last years a relevant number of papers investigated many of them, including sesame seeds as also reported by the authors. In the cited paper (Lastra-Ripoll et al., 2022) extraction, chemical and rheological characterization were performed and also data at different temperatures (even if using frequency sweep tests instead of temperature ramp tests) were obtained.
It could have been more interesting if the authors would have been shown the optimization of the extraction and the possibility to have high yields. Or maybe if this extract could have particularly interesting properties.
In the present form it is an experimental report of some properties of one (among many others) seed extract and it is not suitable for publication in this journal.
Moreover, the paper has many flaws that make questionable some conclusions. The rheological discussion is based on general statements often referred to systems different than that studied in this work. In addition, I suggest a greater attention in manuscript preparation and a language revision.
Here you find further specific comments, I hope that they can be useful to the authors when revising the manuscript before a new submission.
Line 50: the cited reference [1] seems not suitable to give a definition of the term “hydrocolloid”. Different references can be cited. Nevertheless, according to the definition given by the authors (and to other definitions) it seems that the extract obtained by the authors is made by a mixture of hydrocolloids (“… long-chain polymers made of macromolecular proteins and polysaccharides…”), it is not a hydrocolloid. Probably a different name for this extract could be proposed.
Line 54-55: “… hydrocolloids made of proteins frequently display…” “… protein solubility…” I suggest to revise the sentence.
Line 105: “… and air-dried overnight in a hood…” was the adopted system able to control temperature and/or humidity?
Line 106-107: was this the yield of defatting procedure? It should be explained better
Line 110: “… until dissolved completely…”. If the powder is completely dissolved it should be completely soluble and therefore centrifugation should be not necessary… was the powder completely soluble?
Line 124: “… Soxhelt…” should be “… Soxhlet…”
Line 126-127: “… The total carbohydrate content was measured through the subtracting form other compounds…” please revise the sentence
Line 130-139: the paragraph should be completely rewritten explaining how the physical properties were obtained. In this present form it is impossible to understand what the authors did.
Line 145: the authors should explain how they prepared the suspensions for rheological tests.
Line 141-159: performed rheological tests should be described in a more ordered way clearly distinguishing between dynamic tests and flow tests. Test conditions for strain sweep tests, frequency sweep tests and temperature ramp tests should be given. Afterwards test conditions for viscosity determinations can be described.
Line 163: MC301 is a controlled stress rheometer; it can work controlling the strain but it is not a controlled strain rheometer.
Line 170: the terms consistent coefficient and flow index (or flow behavior index) are used for parameters of the power law model describing the viscosity as a function of shear rate. They cannot be used in a model relating G’ and frequency because they are misleading for a reader. Moreover dimension of k’ is wring, it should be Pa.s^q. Parameter q is dimensionless, the symbol (-) should be reported.
Line 191: is extraction yield computed on the basis of original seeds or on the basis of defatted powder?
Line 206-211: the authors are citing literature results (i.e. the presence of simple carbohydrates) obtained from different raw materials and using different extraction conditions. How can they be sure that the same results are valid for their extract? Moreover if they had a large amount of simple carbohydrates instead of long polysaccharides a less structured gel would be expectable.
Line 211-213: the sentence is not clear
Line 214: the meaning of “DI”?
Line 218-219: the sentence is not clear: the insoluble part (i.e. almost 66%) is much greater than protein content (23%), therefore part of carbohydrates is not soluble….
Line 225-227: polysaccharides can contribute increasing viscosity and improving emulsion stability
Line 245: I do not see a “linear” reduction!
Line 251-252: the sentence is misleading, at high shear rate it is known that materials can exhibit a Newtonian plateau in viscosity, they do not behave as Newtonian fluids.
Line 253-257: equation used to fit viscosity data should be clearly reported and parameters should be shown.
Line 258-259: “… which might state that the insoluble aggregates at 3% led to the reduction of viscosity…”. Particles and aggregates should increase viscosity….
Line 260-263: this a speculation without any evidence and misleading. Increasing shear rate cannot induce covalent cross-links. If these cross-links (caused by the higher concentration) are present their effects should be visible also at low shear rates. Moreover, the authors are discussing about small differences among different curves without error bars. Discussed differences could be within the experimental errors. Error bars have to be shown.
Figure 1: viscosity results are surprising when compared to dynamic data. It is unexpected that fluids with viscosity ranging between 1 and 10 mPa.s can exhibit moduli close to 10-30 Pa.s! Are the authors sure about these numbers? According to viscosity data their systems behave more or less like water (having viscosity 1 mPa.s).
Line 269: “Stain” should be “strain”, please check the manuscript the same mistake is present in different parts.
Line 271: when performing dynamic test the rheometer control strain not shear rate…
Line 275-276: a reference for this definition of yield stress is necessary
Line 300-302: It is a speculation without any evidence
Line 334-336: “shear thinning” is the definition commonly used to describe materials having viscosity decreasing with shear rate, it is not related to dynamic tests and frequency dependence
Line 347-350: it seems that the authors neglect the difference between small amplitude oscillations and steady shear tests using the same terms and definitions for both of them. Dynamic tests do not suggest flow behavior (because material is deformed, it doses not flow) and complex viscosity is different with respect to steady shear viscosity.
Line 353-356: the paragraph needs revision. The parameter q=0 indicates G’ constant with frequency, therefore a solid material with a frequency independent behavior. It sis not necessarily a covalent gel….
Line 358-359: shear thinning behavior cannot be related to dynamic tests
Figure 3: in my opinion data at frequencies greater than 2-3 Hz are affected by experimental artefacts due to the inertial effects of rheometer head (see for instance Franck A, Measuring structure of low viscosity fluids in oscillation using rheometers with and without a separate torque transducer, Annual transactions of the Nordic Rheology Society, VOL. 11, 2003)
Line 389-391: the sentence is not clear
Line 394-395: Covalent interactions between protein and polysaccharides? It seems questionable and without any evidence
Abstract and conclusions should be revised and modified according to changes to the manuscript.
Author Response
The paper “Effects of Concentration and Heating/cooling Rate on Rheological Behavior of Sesamum Indicum Seed Hydrocolloid” reports on the chemical and rheological characterization of an extract obtained from sesamum indicum seeds.
The topic is neither particularly interesting nor innovative, different polysaccharides and proteins can be extracted from many vegetable seeds and in the last years a relevant number of papers investigated many of them, including sesame seeds as also reported by the authors. In the cited paper (Lastra-Ripoll et al., 2022) extraction, chemical and rheological characterization was performed and also data at different temperatures (even if using frequency sweep tests instead of temperature ramp tests) were obtained.
It could have been more interesting if the authors would have been shown the optimization of the extraction and the possibility to have high yields. Or maybe if this extract could have particularly interesting properties.
In the present form it is an experimental report of some properties of one (among many others) seed extract and it is not suitable for publication in this journal.
Moreover, the paper has many flaws that make questionable some conclusions. The rheological discussion is based on general statements often referred to systems different than that studied in this work. In addition, I suggest a greater attention in manuscript preparation and a language revision.
Thank you for reviewing the work. As stated in the work by Lastra-Ripoll et al., 2022, they only performed some rheological experiments without attention to the heating and cooling rate on the hydrocolloids structure which definitely change the macromolecules and the application in food products would be changed. They used only some temperatures (10, 25, 40 and 80 oC), however, we consider both heating and cooling rate. Furthermore, the sesame cultivar has also effect on the properties of the macromolecules, although, we did not intend to discuss on it.
C1: Here you find further specific comments; I hope that they can be useful to the authors when revising the manuscript before a new submission.
Line 50: the cited reference [1] seems not suitable to give a definition of the term “hydrocolloid”. Different references can be cited. Nevertheless, according to the definition given by the authors (and to other definitions) it seems that the extract obtained by the authors is made by a mixture of hydrocolloids (“… long-chain polymers made of macromolecular proteins and polysaccharides…”), it is not a hydrocolloid. Probably a different name for this extract could be proposed.
R1: Thanks. It was revised.
C2: Line 54-55: “… hydrocolloids made of proteins frequently display…” “… protein solubility…” I suggest revising the sentence.
R2: Thanks. It was revised.
C3: Line 105: “… and air-dried overnight in a hood…” was the adopted system able to control temperature and/or humidity?
R3: That’s right the temperature and humidity may be changed in a hood, but as mentioned we used oven to dry the seeds and after adding the hexane and removing fat, and air-dried overnight in a hood was used, since the hexane is flammable and hazardous use of oven and furthermore after removing the fat from sesame hydrocolloids, a little amount of hexane was remained in the powder and generally used hood for evaporation of the hexane. This is a general method in oil industry and many papers have used the same procedure, for instance, please see Esmaeili et al., 2015.
C4: Line 106-107: was this the yield of defatting procedure? It should be explained better
R4: That’s right, it is the yield of defatting sesame seed and corrected.
C5: Line 110: “… until dissolved completely…”. If the powder is completely dissolved it should be completely soluble and therefore centrifugation should be not necessary… was the powder completely soluble?
R5: Thank you. The powders are not completely soluble and they are dispersed in water and thus we used the centrifugation.
C6: Line 124: “… Soxhelt…” should be “… Soxhlet…”
R6: Thank, it was corrected.
C7: Line 126-127: “… The total carbohydrate content was measured through the subtracting form other compounds…” please revise the sentence
R7: Thanks, it was corrected.
C8: Line 130-139: the paragraph should be completely rewritten explaining how the physical properties were obtained. In this present form it is impossible to understand what the authors did.
R8: Thanks, it was completely revised.
C9: Line 145: the authors should explain how they prepared the suspensions for rheological tests.
R9: Thank you. It was provided at the end of the section 2.2 and was transferred to the beginning of the section 2.5.
C10: Line 141-159: performed rheological tests should be described in amore ordered way clearly distinguishing between dynamic tests and flow tests. Test conditions for strain sweep tests, frequency sweep tests and temperature ramp tests should be given. Afterwards test conditions for viscosity determinations can be described.
R10: Thanks, All the details in experiments such as the conditions were provided. Firstly the steady state measurement (flow tests) was performed then SAOS measurements, strain sweep, frequency sweep and temperature sweep were done. All the details of the tests were also given.
C11: Line 163: MC301 is a controlled stress rheometer; it can work controlling the strain but it is not a controlled strain rheometer.
R11: That’s right, it was corrected.
C12: Line 170: the terms consistent coefficient and flow index (or flow behavior index) are used for parameters of the power law model describing the viscosity as a function of shear rate. They cannot be used in a model relating G’ and frequency because they are misleading for a reader. Moreover dimension of k’ is wring, it should be Pa.s^q. Parameter q is dimensionless, the symbol (-) should be reported.
R12: The Power-law model for the steady state measurement is different from the Power-law model used in frequency sweep measurements. Please see many papers concerning the use of this model by authors like as:
Hesarinejad, M. A., Jokandan, M. S., Mohammadifar, M. A., Koocheki, A., Razavi, S. M. A., Ale, M. T., & Attar, F. R. (2018). The effects of concentration and heating-cooling rate on rheological properties of Plantago lanceolata seed mucilage. International journal of biological macromolecules, 115, 1260-1266.
Hesarinejad, M. A., Koocheki, A., & Razavi, S. M. A. (2014). Dynamic rheological properties of Lepidium perfoliatum seed gum: Effect of concentration, temperature and heating/cooling rate. Food Hydrocolloids, 35, 583-589.
C13: Line 191: is extraction yield computed on the basis of original seeds or on the basis of defatted powder?
R13: The extraction yield computed on the basis of original seeds.
C14: Line 206-211: the authors are citing literature results (i.e. the presence of simple carbohydrates) obtained from different raw materials and using different extraction conditions. How can they be sure that the same results are valid for their extract? Moreover if they had a large amount of simple carbohydrates instead of long polysaccharides a less structured gel would be expectable.
R14: Thanks for attention. We examined the chemical composition of the sesame hydrocolloids and compare with the other works have been accomplished on sesame proteins or hydrocolloids. Thus, there is a relation between these comparisons. Furthermore, the carbohydrates of the sesame is not simple and mixture of different carbohydrates and if we get insight on the carbohydrate, we should perform HPLC experiments to get more knowledge.
C15: Line 211-213: the sentence is not clear
R15: Thanks, it was corrected.
C16: Line 214: the meaning of “DI”?
R16: Thanks, it was corrected.
C17: Line 218-219: the sentence is not clear: the insoluble part (i.e. almost 66%) is much greater than protein content (23%), therefore part of carbohydrates is not soluble….
R17: That’s right, the carbohydrates have two sections; soluble and insoluble fibre and therefore, the insoluble part is much greater than the protein content.
C18: Line 225-227: polysaccharides can contribute increasing viscosity and improving emulsion stability
R18: Thanks, it was added.
C19: Line 245: I do not see a “linear” reduction!
R19: Thanks, it was corrected.
C20: Line 251-252: the sentence is misleading, at high shear rate it is known that materials can exhibit a Newtonian plateau in viscosity, they do not behave as Newtonian fluids.
R20: OK, it was omitted.
C21: Line 253-257: equation used to fit viscosity data should be clearly reported and parameters should be shown.
R21: OK, it was provided.
C22: Line 258-259: “… which might state that the insoluble aggregates at 3% led to the reduction of viscosity…”. Particles and aggregates should increase viscosity….
R22: Thanks, it was corrected.
C23: Line 260-263: this a speculation without any evidence and misleading. Increasing shear rate cannot induce covalent cross-links. If these cross-links (caused by the higher concentration) are present their effects should be visible also at low shear rates. Moreover, the authors are discussing about small differences among different curves without error bars. Discussed differences could be within the experimental errors. Error bars have to be shown.
R23: Thanks, it was corrected accordingly and omitted the wrong statement.
C24: Figure 1: viscosity results are surprising when compared to dynamic data. It is unexpected that fluids with viscosity ranging between 1 and 10 mPa.s can exhibit moduli close to 10-30 Pa.s! Are the authors sure about these numbers? According to viscosity data their systems behave more or less like water (having viscosity1 mPa.s).
R24: Thanks a lot, and sorry for inconvenience, there is a problem in data calculation, and it was completely revised.
C25: Line 269: “Stain” should be “strain”, please check the manuscript the same mistake is present in different parts.
R25: Thanks a lot, it was corrected.
C26: Line 271: when performing dynamic test the rheometer control strain not shear rate…
R26: Thanks, I am sorry for mistake in writing.
C27: Line 275-276: a reference for this definition of yield stress is necessary
R27: Thanks, a reference was provided.
C28: Line 300-302: It is a speculation without any evidence
R28: Thanks, a reference was provided.
C29: Line 334-336: “shear thinning” is the definition commonly used to describe materials having viscosity decreasing with shear rate, it is not related to dynamic tests and frequency dependence.
R29: Thanks, it was corrected.
C30: Line 347-350: it seems that the authors neglect the difference between small amplitude oscillations and steady shear tests using the same terms and definitions for both of them. Dynamic tests do not suggest flow behavior (because material is deformed, it does not flow) and complex viscosity is different with respect to steady shear viscosity.
R30: Thanks, it was corrected.
C31: Line 353-356: the paragraph needs revision. The parameter q=0 indicates G’ constant with frequency, therefore a solid material with a frequency independent behavior. It is not necessarily a covalent gel….
R31: Thanks, it was corrected.
C32: Line 358-359: shear thinning behavior cannot be related to dynamic tests
R32: Thanks, it was corrected.
C33: Figure 3: in my opinion data at frequencies greater than 2-3 Hz are affected by experimental artefacts due to the inertial effects of rheometer head (see for instance Franck A, Measuring structure of low viscosity fluids in oscillation using rheometers with and without a separate torque transducer, Annual transactions of the NordicRheology Society, VOL. 11, 2003)
R33: Thank you so much, it is completely right, and we only provide the information was achieved by the instrument as stated in the M&M section, the frequency rang 0.1 to 10 was examined and therefore around 2-3 frequency, the moduli values were changed sharply.
C34: Line 389-391: the sentence is not clear.
R34: Thanks, it was corrected.
C35: Line 394-395: Covalent interactions between protein and polysaccharides? It seems questionable and without any evidence
R35: Thanks, the covalent interactions were removed, and it was corrected.
C36: Abstract and conclusions should be revised and modified according to changes to the manuscript.
R36: Thanks, Abstract and conclusions were revised.
Reviewer 3 Report
Manuscript ID: foods-1965981
The article entitled: “Effects of Concentration and Heating/cooling Rate on Rheological Behavior of Sesamum Indicum Seed Hydrocolloid”.
This study was to investigate the effects of concentration and heating/cooling rate on dynamic rheological properties of aqueous dispersion of SISH extracted from sesamum indicum seed under mild conditions. The article is compact and logical. From a methodological point of view, the article uses measurement techniques appropriate to the assumed purpose of the research.
Title
The title corresponds to the content of the article.
Abstract
The abstract includes the aim of the study, methods used in the experiment and contain the principal results and conclusions.
Introduction
Introduction describes the matter of the experiment and determines the examined problem. The authors correctly described the importance of research results. The cited literature refers to the subject of the analyzed problem.
Methods
The data is well collected. In the methods, more details need to be provided (below I have questions). The sampling is appropriate and adequately described. Statistical analysis of measurement results has been used.
2.4. Functional properties of hydrocolloids
Emulsifying properties of the emulsions was measured with a spectrophotometer (absorbance)? Some information would be useful.
The foaming. As above, more information. Reference to literature is not enough.
3. Results and discussion
224: The emulsion stability. Please provide some information about the method in the methods (2.4. Functional properties of hydrocolloids).
391-393: “From Figure 4, it was also observed that the values of G’ and G” kept constant during cooling, indicating that cooling after heating had not significant effect on the storage and loss moduli.” Significantly small changes in value for about 20 minutes, which does not mean that the degree of networking will not change in a longer time.
Figure 4. Proposes the same time scales.
Conclusion
The authors correctly indicate, how the results are related to the studies.
Language
The article is correctly written. English language and style are minor spell check required.
Author Response
C1: Title, The title corresponds to the content of the article.
R1: Thank you.
C2: Abstract, The abstract includes the aim of the study; methods used in the experiment and contain the principal results and conclusions.
R2: Thank you.
C3: Introduction, Introduction describes the matter of the experiment and determines the examined problem. The authors correctly described the importance of research results. The cited literature refers to the subject of the analyzed problem.
R3: Thanks a lot.
C4: Methods, The data is well collected. In the methods, more details need to be provided (below I have questions). The sampling is appropriate and adequately described. Statistical analysis of measurement results has been used.
R4: Thanks a lot.
C5: 2.4. Functional properties of hydrocolloids, emulsifying properties of the emulsions were measured with a spectrophotometer (absorbance)? Some information would be useful.
R5: Thank you, more information about the method of the emulsifying properties was provided.
C6: The foaming. As above, more information. Reference to literature is not enough.
R6: Thank you, more information about the method of the foaming was provided
C7: Results and discussion, the emulsion stability. Please provide some information about the method in the methods (2.4. Functional properties of hydrocolloids).
R7: I appreciated, more information about the method of the emulsifying properties was provided.
C8: 391-393: “From Figure 4, it was also observed that the values of G’ and G” kept constant during cooling, indicating that cooling after heating had not significant effect on the storage and loss moduli.” Significantly small changes in value for about 20 minutes, which does not mean that the degree of networking will not change in a longer time.
R8: Thanks a lot. It was changed accordingly.
C9: Figure 4. Proposes the same time scales.
R9: Thanks a lot. In Fig. 4, a (1oC/min), b (5oC/min), and c (10oC/min), we used different heating/cooling rates and therefore the scale surely changed and cannot be same. If we used the same time scale, the details of the results omitted, and the difference can’t be seen by the readers.
C10: Conclusion, The authors correctly indicate, how the results are related to the studies.
R10: Thanks a lot.
C11: Language, The article is correctly written. English language and style are minor spell check required.
R11: Thanks a lot.
Round 2
Reviewer 2 Report
The manuscript was improved by the authors and in this version the serious flaws of the original submission were partially solved, except for these few issues:
Line 152: the type of mechanical stirrer? (magnetic stirrer, overhead stirrer, rotor stator device?)
Line 180-184 and R12 in authors’ reply: I know very well that power law models can be used to fit both dynamic and steady state data and I know very well differences between power law models used either for steady tests or for dynamic tests. For these reasons I was explaining that the terms “consistency coefficient” and “flow index” are used in power law model for fitting steady shear tests and they cannot be used in power law model for fitting dynamic data (the term flow is meaningless in dynamic tests where deformation is applied, not flow…). Indeed, the authors of cited manuscripts, where the power law models of dynamic tests are used, use only the name “power law parameters”.Please remove these terms when dealing with dynamic data.
Line 211-226 and R14 in authors’ reply: In my opinion it is questionable to “assume” the amount of simple carbohydrates: I agree with the authors about the total amount of polysaccharides but they cannot distinguish between “simple” carbohydrate (such as glucose, arabinose, xylose, galactose, mannose, etc.) and long chain polysaccharides. They do not know which is more abundant.
Line 256-295: I have some concerns about the adopted model: I cannot see a Newtonian plateau at low shear rates (except for two points….) and for parameter n (see Table 2) the error is equal to the parameter suggesting that the parameter is meaningless.
Figure 3 and R33 in authors’ reply: apparently the authors agree with my comment about the experimental artefacts at high frequencies, nevertheless neither the figure was modified, removing the data affected by problems, nor comments about this issue were added to the text. Therefore for a reader not expert in this topic data are correct…. This issue has to be solved by the authors.
Nevertheless, no further information about the novelty or the reasons for this work were given by the authors. Therefore, I have to confirm my opinion that the paper is not suitable for Foods.
Author Response
The manuscript was improved by the authors and in this version the serious flaws of the original submission were partially solved, except for these few issues:
Response: Thank you so much for improving the paper. We have given out the point-by-point responses to your constructive comments as follows.
C1: Line 152: the type of mechanical stirrer? (magnetic stirrer, overhead stirrer, rotor stator device?)
Response: Thank you for your good question. We have added the detailed information about mechanical stirrer in the revised manuscript.
C2: Line 180-184 and R12 in authors’ reply: I know very well that power law models can be used to fit both dynamic and steady state data and I know very well differences between power law models used either for steady tests or for dynamic tests. For these reasons I was explaining that the terms “consistency coefficient” and “flow index” are used in power law model for fitting steady shear tests and they cannot be used in power law model for fitting dynamic data (the term flow is meaningless in dynamic tests where deformation is applied, not flow…). Indeed, the authors of cited manuscripts, where the power law models of dynamic tests are used, use only the name “power law parameters”. Please remove these terms when dealing with dynamic data.
Response: Thank you for your good suggestion. We have removed these terms in the revised manuscript.
C3: Line 211-226 and R14 in authors’ reply: In my opinion it is questionable to “assume” the amount of simple carbohydrates: I agree with the authors about the total amount of polysaccharides but they cannot distinguish between “simple” carbohydrate (such as glucose, arabinose, xylose, galactose, mannose, etc.) and long chain polysaccharides. They do not know which is more abundant.
Response: Thank you for careful comment. Indeed, we just determine the total amount of polysaccharides, but did not perform the NMR test for the breakage of polymeric chain of sesame polysaccharide, and so cannot recognize the monomeric units. It has been revised and removed from the text in the revised manuscript.
C4: Line 256-295: I have some concerns about the adopted model: I cannot see a Newtonian plateau at low shear rates (except for two points….) and for parameter n (see Table 2) the error is equal to the parameter suggesting that the parameter is meaningless.
Response: Thank you for your valuable comment. Since the Carreau model has some problems, it has been removed in the revised manuscript.
C5: Figure 3 and R33 in authors’ reply: apparently the authors agree with my comment about the experimental artefacts at high frequencies, nevertheless neither the figure was modified, removing the data affected by problems, nor comments about this issue were added to the text. Therefore for a reader not expert in this topic data are correct…. This issue has to be solved by the authors.
Response: Thank you for your good comment. We are sorry for the careless revision. We have corrected Figure 3. Also, the methods for the frequency sweep were corrected to frequency 2 Hz.
C6: Nevertheless, no further information about the novelty or the reasons for this work was given by the authors. Therefore, I have to confirm my opinion that the paper is not suitable for Foods.
Response: Thank you for your comment. Actually, to our knowledge, very limited publications focus on the rheological properties of sesame dispersions. Furthermore, no research investigates the rheological properties of SISH as a function of heating/cooling rate at different concentrations. As a novel hydrocolloid, the rheological properties of SISH are of high interest for its widespread usage in the traditional pharmaceutical prescriptions and foods. As a result of diversity in different gum structures and extrinsic conditions within the fluid food systems, the rheological behaviors are different from one gum solution to another. Hence, food companies may have a hard time deciding which gums to use in their fluid food formulations. Thus, an understanding of the rheological properties of SISH is essential for evaluating its potential applications and use as food thickeners or stabilizers. Our results confirmed that SISH can be regarded as an acceptable hydrocolloid for generating natural food components with intriguing functional and rheological qualities in the formulation of microstructured goods.